# *ITGB6*-Knockout Suppresses Cholangiocarcinoma Cell Migration and Invasion with Declining PODXL2 Expression

**DOI:** 10.3390/ijms22126303

**Published:** 2021-06-11

**Authors:** Yurie Soejima, Miho Takeuchi, Nao Miyamoto, Motoji Sawabe, Toshio Fukusato

**Affiliations:** 1Department of Molecular Pathology, Graduate School of Medical and Dental Sciences, Tokyo Medical and Dental University, 1-5-45 Yushima, Bunkyo-ku, Tokyo 113-8510, Japan; apmi56230@gmail.com (M.T.); miyamt.mlab@tmd.ac.jp (N.M.); m.sawabe.mp@tmd.ac.jp (M.S.); 2General Medical Education and Research Center, Teikyo University, 2-11-1 Kaga, Itabashi-ku, Tokyo 178-8605, Japan; fukusato@med.teikyo-u.ac.jp

**Keywords:** cholangiocarcinoma, integrin, *ITGB6*, CRISPR/Cas9, RNA-seq, PODXL2

## Abstract

Intrahepatic cholangiocarcinoma (iCCA) is a heterogeneous bile duct cancer with a poor prognosis. Integrin αvβ6 (β6) has been shown to be upregulated in iCCA and is associated with its subclassification and clinicopathological features. In the present study, two *ITGB6*-knockout HuCCT1 CCA cell lines (*ITGB6*-ko cells) were established using the clustered regulatory interspaced short palindromic repeats (CRISPR), an associated nuclease 9 (Cas9) system, and single-cell cloning. RNA sequencing analysis, real-time polymerase chain reaction (PCR), and immunofluorescent methods were applied to explore possible downstream factors. *ITGB6*-ko cells showed significantly decreased expression of integrin β6 on flow cytometric analysis. Both cell lines exhibited significant inhibition of cell migration and invasion, decreased wound-healing capability, decreased colony formation ability, and cell cycle dysregulation. RNA sequencing and real-time PCR analysis revealed a remarkable decrease in podocalyxin-like protein 2 (*PODXL2*) expression in *ITGB6*-ko cells. Colocalization of PODXL2 and integrin β6 was also observed. S100 calcium-binding protein P and mucin 1, which are associated with CCA subclassification, were downregulated in *ITGB6*-ko cells. These results describe the successful generation of *ITGB6*-ko CCA cell clones with decreased migration and invasion and downregulation of *PODXL2*, suggesting the utility of integrin β6 as a possible therapeutic target or diagnostic marker candidate.

## 1. Introduction

Cholangiocarcinoma (CCA) is a heterogeneous bile duct cancer with poor prognosis, usually classified as intrahepatic, perihilar, or distal, based on its anatomical location [1,2,3,4]. Intrahepatic CCA (iCCA) arises above the second-order bile ducts. iCCA accounts for 10–15% of primary liver cancers and is the second most common primary hepatic malignancy after hepatocellular carcinoma (HCC) [4]. iCCA is an aggressive malignancy with very poor outcomes. Although surgical resection is a potentially curative treatment for early-stage tumors, most patients have advanced disease at diagnosis. The incidence of iCCA has increased globally over the last few decades, whereas the incidences of perihilar CCA and distal CCA have decreased [1,4]. Therefore, iCCA warrants considerable attention.

iCCA can be further subclassified into two main subtypes: large duct type and small duct type [4]. Large duct iCCA arises in the large intrahepatic bile ducts near the hepatic hilus and resembles perihilar and extrahepatic distal CCA, exhibiting macroscopic periductal infiltrating patterns and intraductal growth. Small duct iCCA occurs primarily in the hepatic periphery and has a macroscopic mass-forming growth pattern. These subtypes are evidenced in the immune and molecular features of the tumors and are related to patient prognosis [1,2,3,4].

Recent studies of iCCA suggest the presence of proliferative and inflammatory molecular subclasses that have different clinicopathological features and gene mutations [5], some of which may be candidates for targeted, personalized therapy. Further, there appear to be associations between the inflammation subclass and the large duct type and between the proliferation subclass and the small duct type [4]. However, the precise mechanisms of carcinogenesis and progression in iCCA are still unclear. In addition, because of the heterogeneity of iCCA and the emergence of chemoresistant clones, the development of more effective novel drugs and therapeutic strategies is warranted [6].

Studies from our laboratories have demonstrated that integrin β6 is highly expressed in iCCA, contrasting to its relatively low expression in cholangiolocellular carcinoma (CoCC) and HCC [7]. Integrins are heterodimeric cell surface receptors for cell adhesion and movement found in at least 24 unique combinations of alpha subunits (18 types) and beta subunits (8 types). They are regulated by inside-out signaling across the plasma membrane by cell-extracellular matrix and cell–cell adhesions [8,9]. Integrins are known to be important in cell migration and the regulation of cell proliferation and invasion in cancer [10,11]. Integrin αvβ6 (β6) is expressed exclusively on epithelial cells and is typically expressed only during tissue remodeling, embryogenesis, and carcinogenesis [12]. Integrin β6 is upregulated in several tumor types [13] and promotes invasion in several cancers, including colorectal cancer and pancreatic cancer, through the extracellular signal-regulated kinase and transforming growth factor (TGF)-β pathways that promote matrix metalloproteinase activation [14,15,16]. It has been reported that integrin β6-blocking antibody and antisense integrin β6 treatment suppress tumor growth in animal models, suggesting a possible role for integrin β6 blockade in cancer treatment [16]. In CCA, significant increases in expression of integrins β6 and α6 and their usefulness as diagnostic and prognostic indicators have been reported [17,18,19].

Furthermore, in iCCA, integrin β6 is more highly expressed in the non-peripheral and periductal-infiltrating/intraductal growth types than in the peripheral and mass-forming types, reflecting a close association between integrin β6 expression and the subclassification of iCCA. This might occur mainly through the integrin β6 ligands, tenascin-C, and TGF-β [20]. However, there is currently no direct or mechanistic evidence of the specific role of integrin β6 expression in modifying biological behaviors in iCCA.

The aim of this study was to generate *ITGB6*-knockout (*ITGB6*-ko) iCCA cell clones using clustered regulatory interspaced short palindromic repeats (CRISPR)/CRISPR-associated nuclease 9 (Cas9) gene editing and to confirm the specific effects of integrin β6 on the biological behaviors of iCCA cells. In addition, the molecules correlated with integrin β6 expression were investigated using RNA sequencing (RNA-seq) analysis and real-time polymerase chain reaction (PCR) analysis.

## 2. Results

### 2.1. Establishment of ITGB6-Knockout (ko) Cell Lines

After transfection and single-cell cloning, mutations in the cloned cells were confirmed by direct sequencing and comparison with HuCCT1 wild-type (HuCCT1-wt) cells. A clone with a heterozygous mutation was obtained after the first transfection. This clone was transfected again, which yielded two clones with two homozygous insertion–deletion mutations, *ITGB6*-ko 1 and -ko 2 (Table 1). Premature stop codons were detected in these clones. Integrin β6 protein was significantly reduced in these *ITGB6*-ko cell lines by both immunofluorescence staining (Figure 1A) and flow cytometry (Figure 1B). For flow cytometry, integrin β6 protein expression on the cell membrane and in the cytoplasm was evaluated and described as the mean fluorescence intensity multiplied by the percentage of integrin β6-positive cells (Figure 1C). Integrin β6 expression was markedly reduced in *ITGB6*-ko cells such as HuH28 with a low-*ITGB6* mRNA level compared to HuCCT1-wt cells, indicating the successful establishment of *ITGB6*-ko cell lines.

### 2.2. ITGB6-Knockout Reduced the Migration and Invasion of iCCA Cells

The migration ability of *ITGB6*-ko 1 and -ko 2 cells was significantly reduced compared to that of HuCCT1-wt cells (migrated cells/field: wt, 833.2 ± 57.6; *ITGB6*-ko 1, 206.8 ± 61.3; and *ITGB6*-ko 2, 160.4 ± 11.2; *p* < 0.001) (Figure 2A,B). Similarly, the invasion ability of *ITGB6*-ko cells was also significantly reduced compared with that of HuCCT1-wt cells (invaded cells/field: wt, 589.7 ± 119.8; *ITGB6*-ko 1, 157.6 ± 25.3; and *ITGB6*-ko 2, 144.7 ± 12.7; *p* < 0.001) (Figure 2A,B). In the wound-healing assay, HuCCT1-wt cells migrated further in 96 h than *ITGB6*-ko 1 cells (remaining wound proportion: 0.16 ± 0.074 versus 0.37 ± 0.053, *p* < 0.001). HuCCT1-wt cells also migrated further in 48 h than *ITGB6*-ko 2 cells (remaining wound proportion: 0.48 ± 0.11 versus 0.63 ± 0.11, *p* < 0.01) (Figure 2C,D). These results indicated that the *ITGB6*-knockout significantly inhibited cell migration and invasion, as well as decreased wound-healing capability.

### 2.3. ITGB6-Knockout Inhibited Colony Formation and Induced Cell Cycle Dysregulation in iCCA Cells

The colony formation abilities of *ITGB6*-ko 1 (78 ± 20 × 10^4^ μm^2^) and *ITGB6*-ko 2 (45 ± 8.6 × 10^4^ μm^2^) cells were significantly reduced to 48.9% and 28.3% compared to that of the HuCCT1-wt cells (160 ± 46 × 10^4^ μm^2^) (*p* < 0.05 and 0.01, respectively) (Figure 3A,B), indicating decreased colony formation ability after *ITGB6*-knockout. We assessed cell cycle progression in *ITGB6*-ko cells by flow cytometry. The *ITGB6*-ko 1 and *ITGB6*-ko 2 cells had significantly fewer cells in the G0/G1 fractions than the HuCC1-wt cells (41% and 49%, respectively, versus 63%; *p* < 0.005 for both), and they had significantly more cells in the S/G2/M fractions (43% and 37%, respectively, vs. 31%; *p* < 0.005 and < 0.05, respectively) (Figure 3C,D). Therefore, *ITGB6*-knockout induced cell cycle dysregulation and increased the number of cells in the S/G2 phase. Furthermore, apoptotic responses evaluated by detection of activated caspase-3 were observed in *ITGB6*-ko 1 and *ITGB6*-ko2, while not in HuCCT1-wt (Figure 3E).

### 2.4. RNA-Seq and Real-Time PCR Analysis Revealed a Significant Decrease in PODXL2 Expression in ITGB6-ko Cells

RNA-seq analysis was applied to explore genes associated with *ITGB6*-knockout in iCCA cells. A total of 60,728 genes were expressed in *ITGB6*-ko and HuCCT1-wt cells. Gene ontology and pathway analysis were performed using genes with significantly different expression, i.e., *p* < 0.05 and log2 fold change <−1 (*n* = 479) or log2 fold change >1 (*n* = 621). We focused on assessing the downregulated genes in *ITGB6*-ko 1 cells to explore genes with concordant expression with ITGB gene. In terms of biological process, we found decreased expression of genes involved in cell–cell signaling, extracellular matrix organization, and epithelial cell differentiation (Appendix A). A pathway analysis using the Kyoto Encyclopedia of Genes and Genomes (KEGG) database did not reveal significant pathways involved in migration, invasion, or proliferation (Appendix A). To identify genes closely related to *ITGB6*, we included only those genes with an adjusted *p*-value < 0.05 and a log2 fold change <−1 (*n* = 38) or a log2 fold change >1 (*n* = 92) and extracted genes with a large difference in expression between HuCCT1-wt and *ITGB6*-ko 1 cells (Appendix A). The top 20 downregulated genes in *ITGB6*-ko 1 cells are shown in Table 2. The top 20 upregulated genes in *ITGB6*-ko cells are shown in Table 3.

In addition, to evaluate the possibility of interactive changes in integrin subunit genes, the expression of 35 integrin genes were comparatively examined in the RNA-seq data, but we observed no significant changes other than the slightly decreased expression of integrin subunit alpha E in *ITGB6*-ko 1 cells (data not shown).

We were interested in downregulated genes in knockout cells for therapeutic strategy and selected 5 of the top 20 downregulated genes in *ITGB6*-ko 1 cells, podocalyxin-like 2 (*PODXL2*), claudin 2, S100 calcium-binding protein A2, tetraspanin 8, and galectin 1, which have been associated with cell migration and adhesion in previous studies, and one upregulated gene, CEA cell adhesion molecule 6 (*CEACAM6*), to validate the RNA-seq data using real-time PCR. *PODXL2* had the greatest (negative) fold change and was the most significantly downregulated gene concordantly in both *ITGB6*-ko 1 and *ITGB6*-ko 2 cells compared with HuCCT1-wt cells (*p* < 0.05) (Figure 4), suggesting a possible relationship between *ITGB6* and *PODXL2* expression. *CEACAM6* expression was significantly increased concordantly in both *ITGB6*-ko 1 and *ITGB6*-ko 2 cells compared to HuCCT1-wt cells (*p* < 0.01 and 0.05, respectively) (Figure 4).

### 2.5. Immunofluorescence Staining of PODXL2, CEACAM6, S100P, MUC1, and CD56

Immunofluorescence staining was performed to investigate the protein expression of PODXL2 and CEACAM6 in HuCCT1-wt and *ITGB6*-ko cells. PODXL2 was expressed in the cytoplasm and plasma membrane of HuCCT1-wt cells, and its expression was decreased in *ITGB6*-ko cells (Figure 5A). In addition, it was evident from double immunofluorescence staining that PODXL2 was colocalized with integrin β6 in HuCCT1-wt cells (Figure 5A). CEACAM6 was localized to the cell membrane and cytoplasm, and its expression was higher in *ITGB6*-ko cells than in HuCCT1-wt cells (Figure 5B).

S100 calcium-binding protein P (S100P) and mucin 1(MUC1), which are markers for large bile duct iCCA, were decreased in *ITGB6*-ko 1 and -ko 2 cells compared to HuCCT1-wt cells, whereas CD56 (also called neural cell adhesion molecule), a marker for small bile duct iCCA, was slightly increased in *ITGB6*-ko 1 (Figure 6). These results suggested that proteins that are predominant in large bile duct iCCA are downregulated in *ITGB6*-ko cells.

### 2.6. Relationship between PODXL2 Expression and Clinicopathological Features in Patients with Intrahepatic Cholangiocarcinoma (iCCA)

Positive immunohistochemical staining for PODXL2 were observed in 17 (32.7%) of 52 iCCAs. PODXL2 expression was localized in the cell membrane and cytoplasm. The correlations between PODXL2 expression and the localization, histological differentiation, growth type, serosa invasion, and bile duct invasion of iCCA were statistically significant (Table 4). PODXL2 expression was higher in the non-peripheral central localization type iCCA than in the peripheral localization type (*p* = 0.0197); lower in the poorly differentiated type than in the well-differentiated type (*p* = 0.0104); and higher in the infiltrative-growth type than in the expansive-growth type (*p* = 0.0223). High expression was related to serosa invasion and bile duct invasion (*p* = 0.0239 and 0.0038, respectively).

### 2.7. Correlation between Integrin β6 Expression and PODXL2 Expression in iCCA Tissues

The expression of integrin β6 was observed in 31 (59.6%) of iCCA. In this study, integrin β6 expression was considered positive when more than 5% of the tumor cells or areas showed positive staining. The localization of integrin β6 expression was in cell membrane, cytoplasm, and basal lamina. Integrin β6 expression was associated with macroscopic type (*p* = 0.0362), intrahepatic metastasis (*p* = 0.0417), and lymph node metastasis (*p* = 0.0194) (Appendix A). Furthermore, PODXL2 expression was significantly correlated with integrin β6 expression within tumors (*r* = 0.569, *p* = 0.000049) (Figure 7).

## 3. Discussion

In this study, *ITGB6*-ko iCCA cell lines were generated using the CRISPR/Cas9 system and single-cell cloning. Direct sequence analysis after transfection of gRNAs into HuCCT1 cells showed a single-base insertion in the HuCCT1 *ITGB6*-ko 1 line and a single-base insertion and large deletion in the HuCCT1 *ITGB6*-ko 2 line. Since these two clones do not have the original sequence of *ITGB6*, we regarded them as *ITGB6*-ko cell lines with homozygous mutations and a premature stop codon, although the effects of a large deletion in *ITGB6*-ko 2 line was not evaluated sufficiently. To confirm this, we performed immunofluorescence staining and flow cytometry, which indicated significantly decreased expression of integrin β6 protein in these cell lines compared with the HuCCT1-wt cells. Integrin β6 is the rate-limiting subunit for αvβ6 heterodimer formation. The expression of integrin αvβ6 is limited by the transcription of *ITGB6*, which is regulated by promotor-binding sites and phosphorylation of transcriptional factors, including signal transducer and activator of transcription (STAT) 3, CCAAT enhancer-binding protein alpha (C/EBPα), ETS proto-oncogene 1 transcription factor (ETS1), and SMAD family member 3 (SMAD3). In our preliminary study using iCCA cell lines with significantly high and low integrin β6 expression (HuCCT1 and HuH28, respectively), no significant mutations in *ITGB6* or its promoter, except for a few silent mutations, were detected by direct sequencing analysis (unpublished data), suggesting that integrin β6 expression is not regulated by *ITGB6* gene mutation but by transcription.

After establishing these *ITGB6*-ko cell lines, the effects of the *ITGB6*-knockout on the biological behaviors and functions of iCCA cells were evaluated. Cell migration and invasion were markedly decreased in *ITGB6*-ko cells (*ITGB6*-ko 1 and -ko 2) in transwell assays and wound-healing assays when compared with HuCCT1-wt cells. Colony formation ability was also reduced in both knockout cell lines. In a previous study using integrin β6-specific siRNA in CCA cells (RBE and QBC939), cell migration, invasion, and proliferation were significantly suppressed by integrin β6 silencing [18]. *ITGB6*-knockout in this study also inhibited cell migration and invasion, decreased colony formation, and affected cell cycle progression and apoptosis, indicating the close involvement of integrin αvβ6 in the regulation of migration, invasion, and proliferation in iCCA, as has been previously observed in other cancers [15,16].

We previously reported that integrin β6 was significantly downregulated in CoCC of low-grade malignant potential but upregulated in iCCA of aggressive malignant potential [7]. In addition, our previous study revealed the increased expression of integrin β6 in the central large bile duct and periductal infiltrating subtypes of iCCA, indicating a close relationship with iCCA subclassification [20]. In the present study, S100P and MUC1, proteins that are highly expressed in large bile duct iCCA [21,22], were decreased in the *ITGB6*-ko cell lines. These results suggest that HuCCT1 cells may have the characteristics of large bile duct iCCA but transform into small bile duct iCCA-like cells after integrin β6 suppression. The phenotypic change between large duct type and small duct type is interesting because previous studies have shown that small duct iCCA has a higher 5-year postoperative survival rate than large duct iCCA [4,21]. As the iCCA subtypes are based on mucin productivity and immunophenotype [4,22], our results suggest that integrin β6 might be a useful target for therapeutic and diagnostic strategies.

After we observed that *ITGB6*-knockout induced profound effects on iCCA cells and modified their biological behaviors, we wanted to determine the precise molecular pathways affected by integrin β6 and its specific effects in these cell lines. Therefore, we investigated the differentially regulated genes in *ITGB6*-ko cells using RNA-seq and real-time PCR analysis. The results showed a remarkable decline in *PODXL2* expression in both *ITGB6*-ko cell lines compared to the wild-type HuCCT1 cells. PODXL2 is a newly discovered member of the CD34 family and a type-I transmembrane sialomucin that functions as an L-selectin ligand [23,24,25]. The role of PODXL2 in cancer is not yet known, and no interacting partners or regulators have been identified. However, PODXL1, another CD34 family member, is overexpressed in several cancers, and high PODXL1 is closely associated with a poor prognosis in cancer patients [26,27]. In the present study, we demonstrated a possible relationship between PODXL2 and integrin β6. A significant decrease in PODXL2 expression might be associated with decreased migration and invasion, as well as decreased colony formation, in *ITGB6*-ko iCCA cells. In BT474 breast cancer cells, knockdown of PODXL2 slightly suppressed tumor cell migration and reduced expression of cancer stem cell markers through the Rac family small GTPase 1 pathway [28]. Inhibition of colony formation in *ITGB6*-ko iCCA cells in the present study might also be induced by reduced expression of PODXL2, because PODXL2 might be necessary for the maintenance of clonogenic potential in CCA cells as indicated in breast cancer cells. In addition, it is interesting that PODXL2 was colocalized with integrin β6 in HuCCT1 cells, suggesting a close association between these two adhesion molecules. Furthermore, the relationship between PODXL2 expression and clinicopathological features in iCCA patients and significant correlation between integrin β6 expression and PODXL2 expression in iCCA tissues was also evident in the present immunohistochemical studies.

CRISPR/Cas9 is widely used as a gene-editing method to investigate the molecular basis of cancers [29,30]. The gRNA sequence has a crucial role in the efficiency and specificity of CRISPR/Cas9 by reducing off-target effects [31]. In the present study, we used this novel, powerful gene-editing tool to explore new molecules in the integrin pathway within iCCA cells and observed a close association between integrin β6 and PODXL2, which was first described in the present study. However, the precise role of PODXL2 in the integrin pathway is still unknown, so further studies evaluating the transcriptional regulation of both *ITGB6* and *PODXL2* through phosphorylation of transcription factors such as STAT3, C/EBPα, and SMAD3 in *ITGB6*-ko iCCA cells are warranted. In addition, the effect of integrin β6 on the invasive growth of iCCA cells in vivo has not yet been evaluated. In future studies, we want to establish a xenograft tumor model in nude mice bearing *ITGB6*-ko iCCA cells.

The present study is the first to successfully generate *ITGB6*-ko iCCA cell clones, which had decreased migration and invasion in vitro, as well as downregulation of PODXL2, suggesting the utility of integrin β6 as a possible therapeutic or diagnostic candidate. Because of its epithelium-specificity and its dependency on specific pathological conditions, integrin β6 expression has high affinity and de novo expression in cancer tissues and represents a promising cancer cell target [14]. We were interested in downregulated genes in knockout cells for therapeutic strategy. Application of integrin β6-specific function blocking monoclonal antibodies and nonpeptidic and peptidic integrin ligands with antagonizing effects has been considered as a potential therapeutic approach [14,16,32]. Furthermore, imaging for cancers using novel integrin β6 ligands encourages a promising tool for cancer diagnosis [14,33].

## 4. Materials and Methods

### 4.1. Cell Line and Cell Culture

The human CCA cell line, HuCCT1 [34] and HuH28 [35] (Health Science Research Resources Bank, Osaka, Japan), which have significantly high and low expression of *ITGB6* mRNA, respectively [7], was maintained in RPMI-1640 medium (Sigma–Aldrich, St. Louis, MO, USA) supplemented with 10% fetal bovine serum (FBS) (Sigma–Aldrich) and 1% antibiotic-antimycotic (Gibco, Dublin, Ireland) in a humidified atmosphere at 37 °C with 5% CO_2_. These two cell lines were selected after previous experiments using five CCA cell lines with different characteristics reflecting heterogeneous cancer [7].

### 4.2. Patients and Tissue Samples

Tissue samples from 52 patients with iCCA were obtained by surgical resection at Tokyo Medical and Dental University Hospital between 2007–2018. Tissue samples were fixed in a 10% formalin solution, and embedded in paraffin for histological diagnosis and immunohistochemistry analysis. The patients included 39 men and 13 women ranging from 39 to 84 years of age (mean, 70.8 years). Six patients were positive for the serum hepatitis B surface antigen (HBsAg), and eight were positive for the anti-hepatitis C virus (HCV) antibody; one was positive for both, and 39 were negative for both. The largest tumor diameters ranged from 18 to 220 mm (mean, 58.9 mm). iCCA was grossly classified into three groups: mass-forming (MF), periductal-infiltrating (PI), and intraductal-growth (IG), as well as mixed types (MF + PI and IG + PI). Based on surgical findings and macroscopic examination, the tumors were defined as peripheral localization types involving the septal and interlobular bile ducts, or as non-peripheral central localization types involving the first branches of the right and left hepatic bile ducts. The histological differentiation grades for iCCA were assigned according to World Health Organization classifications [4]. The surrounding non-tumorous liver tissues showed normal liver in 33 patients, chronic hepatitis in 12 patients, and cirrhotic change in seven patients. The diagnosis of iCCA was confirmed by immunohistochemical results that were positive for cytokeratin 7 and negative for hepatocyte marker (Hep Par-1), in addition to the histological findings. This study was approved by the ethics committee of Tokyo Medical and Dental University (No. M2000-2081; 8 May 2015).

### 4.3. Transfection and Genomic Cleavage Detection

HuCCT1 cells (5 × 10^4^ cells) with high expression of ITGB6 mRNA were cultured in 24-well plates to subconfluency, transfected with guideRNA (gRNA; #1, #2, or #3), and TrueCut Cas9 Protein v2 (Invitrogen, Carlsbad, CA, USA) using Lipofectamine CRISPRMAX Cas9 Transfection Reagent (Invitrogen) in accordance with the manufacturer’s protocols, and cultured for 48 h. gRNAs (TrueGuide Synthetic gRNAs, Invitrogen) against *ITGB6* were designed to be complementary to three target DNA sequences (#1, #2, and #3) (Figure 8). Genomic DNA cleavage efficiency was analyzed using the Gene Art Genomic Cleavage Detection Kit (Invitrogen). Briefly, primers for PCR amplification were designed (Appendix A), and DNA extraction and PCR amplification of target sequences were followed by enzyme treatment, in accordance with the manufacturer’s protocol. After agarose gel electrophoresis, each band pattern was evaluated using image analysis software (ImageJ, National Institute of Health, Bethesda, MD, USA) to calculate the cleavage efficiency. Finally, transfection for gene knockout was performed in a 6-well plate (2.5 × 10^5^ cells) using the gRNA #2 with the highest cleavage efficiency. gRNA #1 and #3 with low cleavage efficiency were not used in the following transfection experiment for gene knockout.

### 4.4. Single-Cell Cloning and Direct Sequence Analysis

After transfection, HuCCT1 cells were seeded into a 96-well plate at a concentration of 0.5 cells/well. Single cells were identified under the microscope and subcultured into 24-well plates after single-cell outgrowth. DNA was extracted from each monoclonal line using a QIAamp DNA Mini Kit (Qiagen, Hilden, Germany). After PCR, the amplicons of the target sequences were directly sequenced (FASMAC, Atsugi, Japan) to determine mutations. Sequencing was performed after TA cloning (TOPO TA Cloning Kit; Invitrogen) for some samples.

### 4.5. Immunofluorescence Staining

Immunofluorescence staining was used to verify whether integrin β6 protein was reduced in the *ITGB6*-ko HuCCT1 clones and to investigate the expression of S100P, MUC1, CD56, PODXL2, CEACAM6, and cleaved caspase-3 in these clones. HuCCT1-wt cells, cells transfected with *ITGB6* gRNAs (*ITGB6*-ko) and HuH28 (5 × 10^4^ cells) were cultured in 4-well chamber slides (IWAKI, Tokyo, Japan) to subconfluency and fixed in 4% paraformaldehyde/phosphate-buffered saline (PFA/PBS), followed by permeabilization with 0.1% Triton X-100/PBS for 10 min. After treatment with C-Block (casein-blocking solution; Genemed Biotechnologies, South San Francisco, CA, USA) for 20 min at room temperature, cells were incubated with mouse anti-integrin β6 (1:100; Merck, Darmstadt, Germany), rabbit anti-S100P (1:200; Abcam, Cambridge, UK), mouse anti-MUC1 (1:100; Novocastra Laboratories, Newcastle upon Tyne, UK), mouse anti-CD56 (1:100; Novocastra), rabbit anti-PODXL2 (1:200; Atlas Antibodies, Bromma, Sweden), rabbit anti-CEACAM6 (1:500; Abcam), or rabbit anti-cleaved caspase-3 (Asp175) (1:400; Cell signaling Technology, Danvers, MA, USA) antibodies overnight at 4 °C. After washing with 0.1% Tween 20/PBS, the cells were incubated with fluorescence-labeled secondary antibodies (goat anti-mouse IgG H&L, Alexa Fluor 594) goat anti-rabbit IgG H&L, Alexa Fluor 488; Abcam, diluted 1:1000) for 1 h at room temperature. Nuclei were stained with DAPI (Invitrogen). Stained cells were mounted with a fluorescence mounting medium (DAKO, Glostrup, Denmark) and examined using a Keyence BZ-X700 microscope (Keyence, Osaka, Japan). Double immunofluorescence staining for integrin β6 and PODXL2 was also performed, and the merged images were observed to assess colocalization.

### 4.6. Flow Cytometry

We also confirmed the expression of integrin β6 in HuCCT1-wt and *ITGB6*-ko cells by flow cytometry as well as HuH28 cells as a control. Both membranous and intracellular integrin β6 were examined. Cells (1 × 10^5^–10^6^) were collected and fixed with 4% PFA/PBS at room temperature for 15 min. After washing with 1% FBS/PBS, cell membranes were permeabilized with 0.1% Triton-X/PBS at room temperature for 10 min. Cells were incubated with mouse monoclonal anti-integrin β6 (1:100; Merck) as the primary antibody and mouse IgG1 (1:100; DAKO) as an isotype control for 30 min at 4 °C. After washing with 1% FBS/PBS, cells were incubated with fluorescently labeled goat anti-mouse IgG H&L antibody (Alexa Fluor 488, Abcam, 1:2000) for 30 min at 4 °C, and then the stained cells were analyzed by flow cytometry (CytoFLEX, BECKMAN COULTER, Brea, CA, USA).

### 4.7. In Vitro Transwell Migration and Invasion Assays

Five hundred microliters of serum-free RPMI-1640 medium were placed into Matrigel-coated inserts and wells of the invasion chambers (Corning, NY, USA), which were placed at 37 °C for 2 h. The hydrated Matrigel-coated inserts for the invasion assay and the non-coated inserts for the migration assay were placed into 24-well plates containing 750 μL of RPMI-1640 medium with 10% FBS. HuCCT1 cells (wt or *ITGB6*-ko) (5 × 10^5^ cells) in serum-free medium were seeded into the insert. After incubation for 24 h at 37 °C, the cells were fixed and stained with Diff-Quik solution (Sysmex, Hyogo, Japan). The membranes were then removed and mounted on a slide. The numbers of migrated and invaded cells were counted in five random fields under a light microscope (10× magnification), and the average value of the five fields was calculated.

### 4.8. Wound-Healing Assay

HuCCT1 cells (wt or *ITGB6*-ko) (5 × 10^4^ cells) were plated into 24-well plates and incubated for 72 h to reach confluence. After replacing the media with serum-free media, the cell monolayer was scratched with a sterile pipette tip and photographed under a light microscope at 0, 24, 48, 72, 96, and 120 h. The area of the wound was measured, and the migration rate was calculated based on the width of the 0-h timepoint (BZ-X700 Analyzer, Keyence).

### 4.9. Colony Formation Assay

HuCCT1 cells (wt or *ITGB6*-ko) (5 × 10^2^ cells) were seeded into 6-well plates and cultured (37 °C, 5% CO_2_) for 10 days. The cell colonies were fixed with methanol for 30 min and then stained with 0.1% crystal violet for 30 min. Five random fields were photographed under a light microscope (10× magnification), and the area of the colonies was measured using an image analyzer (BZ-X700 Analyzer, Keyence).

### 4.10. Cell Cycle Analysis

Cells (1 × 10^5^–10^6^) were trypsinized and fixed with 70% ethanol at 4 °C for at least 30 min. After washing with PBS, 50 μL of 100 μg/mL of RNase solution was added, followed by incubation at 37 °C for 20 min. Next, 25 μg/mL of propidium iodide solution was added, followed by incubation in the dark at 4 °C for 30 min, and then the cell cycle distribution was assessed in triplicate by flow cytometry (CytoFLEX, BECKMAN COULTER).

### 4.11. RNA-Seq Analysis

RNA-seq was performed on *ITGB6*-ko 1 with single-base insertion. Total RNA was extracted from HuCCT1-wt and *ITGB6*-ko 1 cells (*n* = 3 each) using a RNeasy Mini Kit (Qiagen). RNA-seq was performed by Takara Bio Inc. (Shiga, Japan). Sequencing libraries were constructed using the SMART-Seq v4 Ultra Low Input RNA Kit (Clontech Laboratories, Mountain View, CA, USA) for sequencing and the Nextera XT DNA Library Prep Kit and Nextera XT Index Kit v2 SetA/B/C/D (Illumina, San Diego, CA, USA) for libraries. The libraries were sequenced on a NovaSeq 6000 system (Illumina) and mapped to the human genome GRCh38 using the DRAGEN Bio-IT Platform (Illumina). Differentially expressed genes were identified using a *t*-test (*p* < 0.05) and log fold change (log2 <−1 or >1). To interpret the biological functions of the differentially expressed genes, gene ontology and pathway analysis were performed using DAVID v6.8 (National Institute of Allergy and Infectious Diseases) and the KEGG database. We then narrowed down the target genes using the adjusted *p*-value and log fold change (log2 < −1 or >1) after the application of the Benjamini–Hochberg procedure to decrease the number of false positives.

### 4.12. Real-Time PCR

Total RNA was extracted from HuCCT1-wt, *ITGB6*-ko 1, and *ITGB6*-ko 2 cells using an RNeasy Mini Kit (Qiagen) in accordance with the manufacturer’s protocol. One microgram of total RNA was reverse-transcribed to cDNA using a QuantiTect RT Kit (Qiagen) with oligo(dT) and random hexamers in accordance with the manufacturer’s instructions. Quantitative PCR was performed using Power SYBR Green Master Mix (Thermo Fisher Scientific, Waltham, MA, USA) and custom-made primers. The PCR primers used in this study were synthesized by FASMAC (Appendix A). Expression was normalized to that of glyceraldehyde-3-phosphate dehydrogenase. All reactions were performed in triplicate using the ABI 7900HT real-time PCR system (Applied Biosystems, Foster City, CA, USA).

### 4.13. Immunohistochemistry and Evaluation

Paraffin-embedded tissue sections that were 4-μm in thickness were deparaffinized with xylene and rehydrated with graded ethanol. Antigen retrieval was performed with pH6 citrate buffer for PODXL2, and Proteinase K for integrin β6. Endogenous peroxidase was quenched with 3% H_2_O_2_ in distilled water for 5 min. The slides were incubated with rabbit anti-PODXL2 antibody (1:300; Atlas Antibodies) and mouse monoclonal anti-integrin β6 (1:400; Merck) for 30 min at room temperature, followed by incubation with the secondary antibody and detection using a Histofine Simple Stain MAX PO (MULTI) (Nichirei Biosciences, Tokyo, Japan) in accordance with the manufacturer’s protocol, and counterstained with hematoxylin. PODXL2 and integrin β6 expressions were considered positive when more than 5% of the tumor cells or areas showed positive staining regardless of the intensity.

### 4.14. Statistical Analysis

Each experiment was carried out in triplicate, and the data were expressed as the mean ± standard deviation. The data of the Transwell assays, colony formation assays, and wound-healing assays were assessed using analysis of variance (Tukey), and mRNA levels were assessed using one-way analysis of variance (Dunnett). The correlations between the clinicopathological findings and PODXL2 or integrin β6 expression in iCCA were assessed using Fisher’s exact or χ^2^ and Student’s *t*-tests. The correlation between PODXL2 expression and integrin β6 expression in iCCA was analyzed based on Spearman’s rank correlation coefficient. A *p* < 0.05 was considered significant.

## Figures and Tables

**Figure 1 ijms-22-06303-f001:**
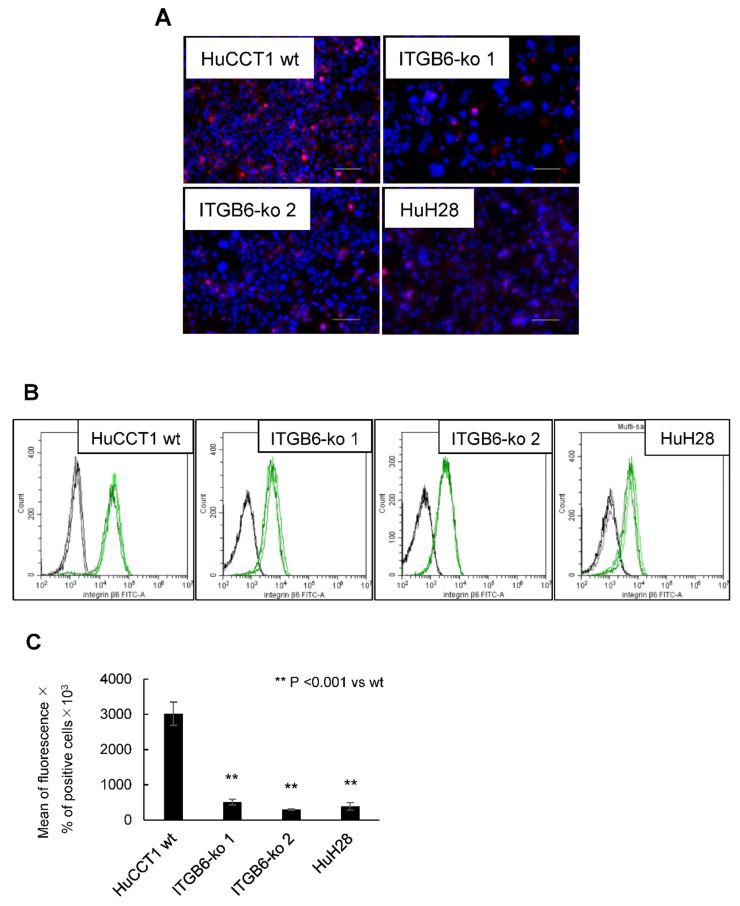
Integrin β6 protein was reduced in integrin β6-knockout (*ITGB6*-ko) cell lines by immunofluorescence staining and flow cytometry. (**A**) Integrin β6 staining (red) in the cytoplasm and plasma membrane of HuCCT1 wild-type (wt), *ITGB6*-ko cells, and HuH28 (scale bar, 100 μm). (**B**) Representative histograms for integrin β6 expression determined by flow cytometry (black lines, mouse IgG1; green lines, mouse monoclonal anti-integrin β6). (**C**) The mean fluorescence intensity was multiplied by the percentage of integrin β6-positive cells to determine integrin β6 expression. The expression of integrin β6 was significantly decreased in *ITGB6*-ko 1 and -ko 2 cells compared with HuCCT1-wt cells (*p* < 0.001) as well as HuH28 cells as a negative control. Experiments were performed in triplicate.

**Figure 2 ijms-22-06303-f002:**
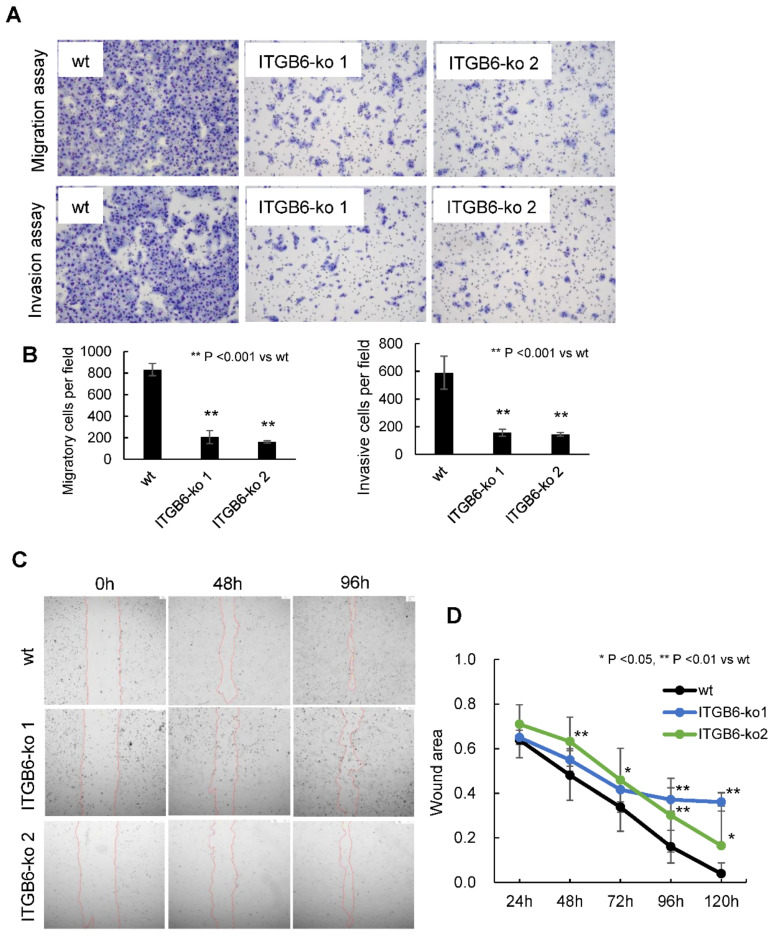
Integrin β6-knockout (*ITGB6*-ko) cells have decreased patterns of migration and invasion. (**A**) Migratory and invasive cell were stained with Diff-Quik solution. (**B**) Migratory cells per field and invasive cells per field were significantly decreased in *ITGB6*-ko cells compared to wild-type (wt) cells (*p* < 0.001 for both). The results shown represent the mean ± standard deviation from three independent experiments. (**C**) Wound closure of HuCCT1-wt, *ITGB6*-ko 1, and *ITGB6*-ko 2 cells after scratching. (**D**) Quantitation of wound areas represent the results of three independent experiments. *ITGB6*-ko 1 and -ko 2 cells were significantly less migratory than wild-type cells.

**Figure 3 ijms-22-06303-f003:**
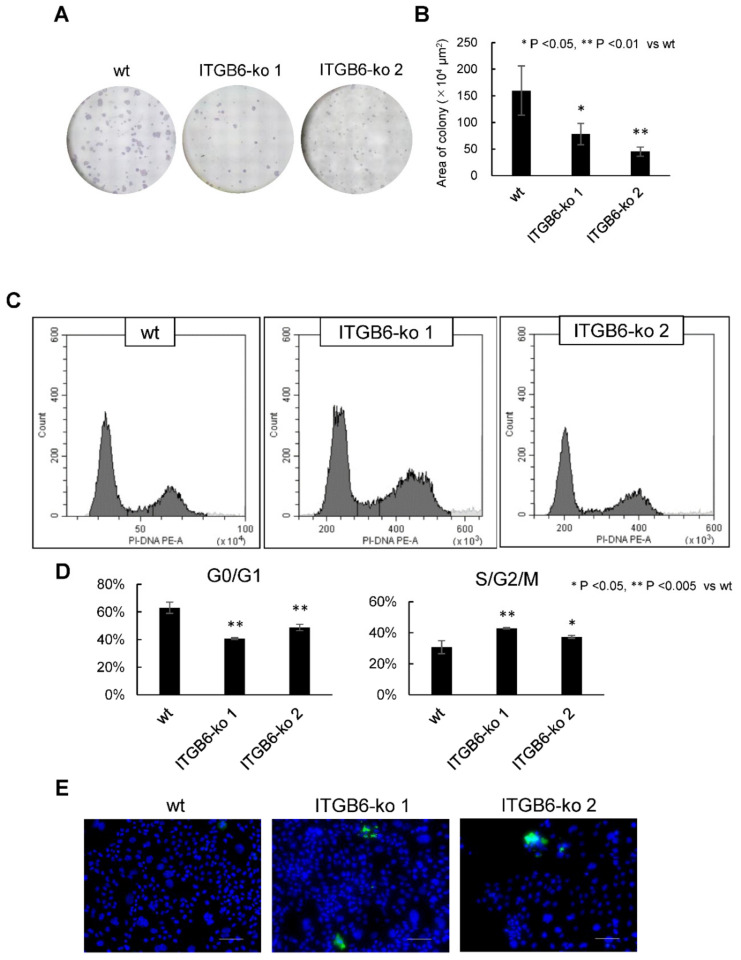
Integrin β6-knockout (*ITGB6*-ko) affected colony formation and cell cycle progression. (**A**) Colonies were stained with crystal violet. (**B**) The *ITGB6*-ko cells colonies covered significantly less surface area than the wild-type (wt) cell colonies. (**C**) Representative histograms of cell distribution in each cell cycle phase determined by flow cytometry. (**D**) The number of cells in G0/G1 was lower in *ITGB6*-ko cells than in wt cells, whereas the number of cells in S/G2/M was increased in *ITGB6*-ko cells. All experiments were performed in triplicate. (**E**). Cleaved caspase-3 immunofluorescence was observed in *ITGB6*-ko cells, while not in wild-type cells. DAPI was used to stain nucleus. Green and blue signals indicate cleaved caspase-3 and nucleus respectively.

**Figure 4 ijms-22-06303-f004:**
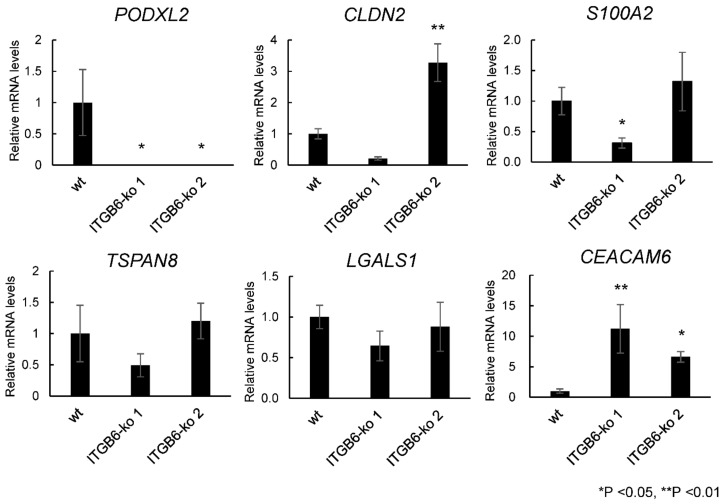
Real-time polymerase chain reaction analysis. mRNA levels of podocalyxin-like 2 (*PODXL2*) were significantly decreased concordantly in both integrin β6-knockout (ko) 1 and -ko 2 cells compared with HuCCT1 wild-type cells, whereas CEA cell adhesion molecule 6 (*CEACAM6*) mRNA levels were increased concordantly. The mRNA level of claudin 2 was significantly increased in *ITGB6*-ko 2 cells, and S100 calcium-binding protein A2 mRNA level was significantly decreased in *ITGB6*-ko 1 while tetraspanin 8 and galectin 1 showed no significant change. All experiments were performed in triplicate.

**Figure 5 ijms-22-06303-f005:**
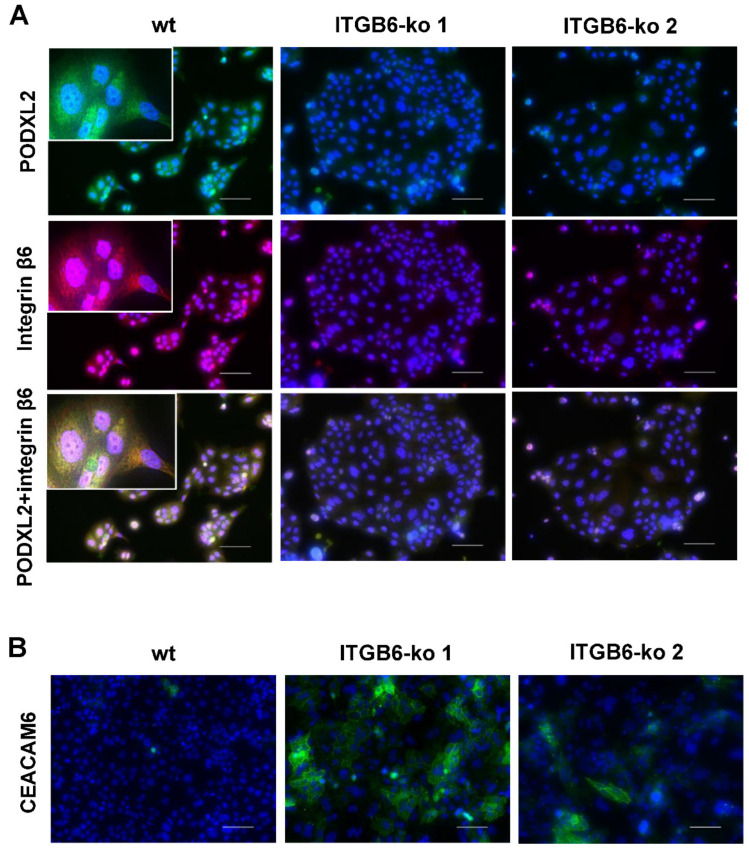
Immunofluorescence staining of podocalyxin-like protein 2 (PODXL2), integrin β6, and CEA cell adhesion molecule 6 (CEACAM6) in HuCCT1 wild-type (wt) and integrin β6-knockout (*ITGB6*-ko) cells. (**A**) PODXL2 (green) was strongly expressed in HuCCT1-wt cells and significantly downregulated in ITGB6-ko cells. The expression of integrin β6 is shown in red. Merged images of PODXL2 and integrin β6 are yellow (inset), indicating colocalization of PODXL2 and integrin β6. (**B**) CEACAM6 was localized to the cell membrane and cytoplasm and was increased in *ITGB6*-ko cells compared with wild-type cells. (Scale bar, 100 μm).

**Figure 6 ijms-22-06303-f006:**
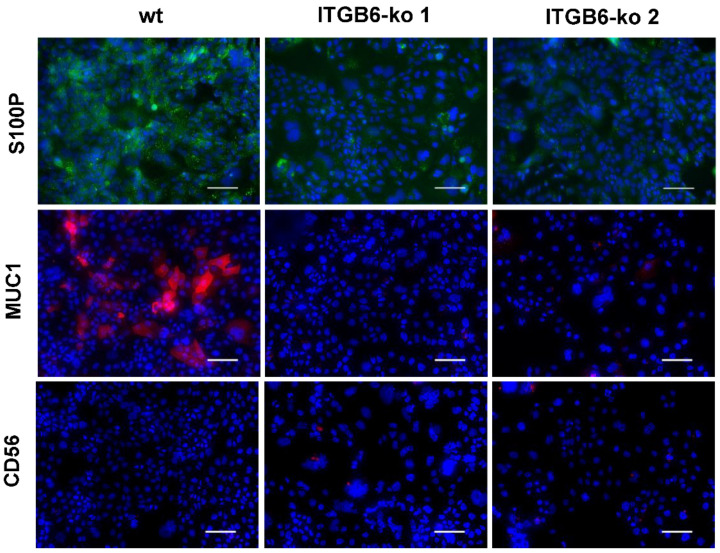
Immunofluorescence staining of S100 calcium-binding protein P (S100P), mucin 1 (MUC1), and CD56 in HuCCT1 wild-type (wt) and integrin β6-knockout (*ITGB6*-ko) cells. The expressions of S100P (green) and MUC1 (red) decreased in *ITGB6*-ko cells compared with wild-type cells, whereas the expression of CD56 (red) showed slight increase in *ITGB6*-ko 1. (Scale bar, 100 μm).

**Figure 7 ijms-22-06303-f007:**
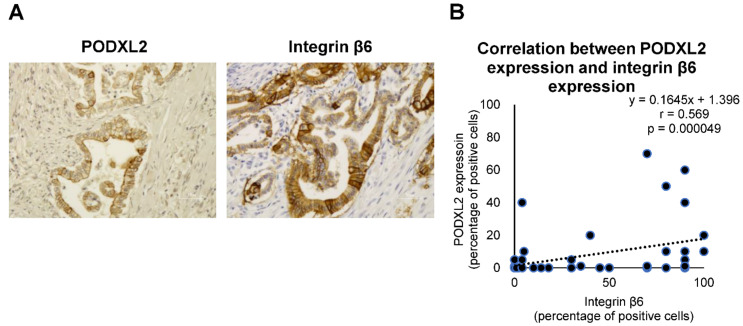
Expressions of PODXL2 and integrin β6 in intrahepatic cholangiocarcinoma tissue. (**A**) PODXL2 and integrin β6 are shown in the cell membrane and cytoplasm of tumor cells. PODXL2 expression patterns resemble integrin β6 expression patterns. Scale bar, 50 μm. (**B**) Correlation between PODXL2 expression and integrin β6 expression within tumors was significant (*r* = 0.569, *p* = 0.000049).

**Figure 8 ijms-22-06303-f008:**
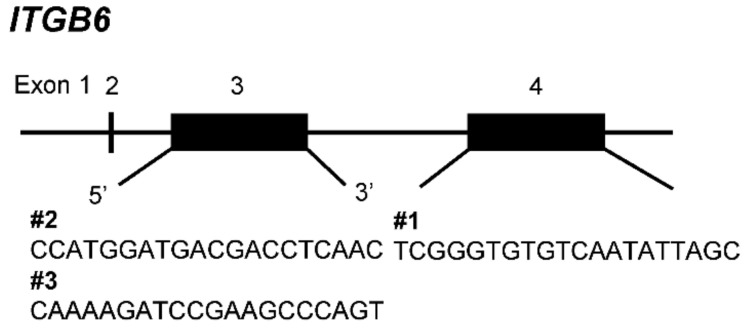
guideRNAs were designed complementary to target DNA sequences (#1–3) in exon 3 and exon 4 of integrin β6 (*ITGB6*).

**Table 1 ijms-22-06303-t001:** Direct sequence analysis of target genes in *ITGB6*-knockout monoclonal cells.

HuCCT1	Sequence of Target Genes	Results	Amino Acid
wt	(T)CCATGGATGACGACCTCAAC		SMDDDLN
ITGB6-ko 1	(T)CCA**T**TGGATGACGACCTCAAC	+1	SIG(STOP)
ITGB6-ko 2	(T)CCA**T**TGGATGACGACCTCAAC	+1	SIG(STOP)
	ATG**----------**ACCTCAACACAATAAAGG	−25	MTSTQ(STOP)

wt: wild-type, ITGB6-ko: integrin β6-knockout, stop codon: TAA; TAG; TGA.

**Table 2 ijms-22-06303-t002:** Top 20 downregulated genes in *ITGB6*-knockout cholangiocarcinoma (HuCCT1) cell line.

Entrez Gene ID	Gene Symbol	Gene Name	Log2 Fold Change	Adjusted *p*-Value
50512	PODXL2	podocalyxin-like 2	−7.50	0.021
5915	RARB	retinoic acid receptor beta	−4.42	0.015
93659	CGB5	chorionic gonadotropin subunit beta 5	−2.91	0.017
23584	VSIG2	V-set and immunoglobulin domain-containing 2	−2.88	0.023
9075	CLDN2	claudin 2	−2.82	0.021
79966	SCD5	stearoyl-CoA desaturase 5	−2.59	0.045
4495	MT1G	metallothionein 1G	−2.42	0.045
6273	S100A2	S100 calcium-binding protein A2	−2.23	0.025
9022	CLIC3	chloride intracellular channel 3	−1.89	0.017
725	C4BPB	complement component 4-binding protein beta	−1.83	0.045
56925	LXN	latexin	−1.74	0.039
7103	TSPAN8	tetraspanin 8	−1.53	0.042
5054	SERPINE1	serpin family E member 1	−1.51	0.023
3956	LGALS1	galectin 1	−1.35	0.017
8263	F8A1	coagulation factor VIII-associated 1	−1.34	0.026
230	ALDOC	aldolase, fructose-bisphosphate C	−1.21	0.019
1075	CTSC	cathepsin C	−1.15	0.018
6319	SCD	stearoyl-CoA desaturase	−1.12	0.007
3149	HMGB3	high mobility group box 3	−1.08	0.021
39	ACAT2	acetyl-CoA acetyltransferase 2	−1.03	0.017

**Table 3 ijms-22-06303-t003:** Top 20 upregulated genes in *ITGB6*-knockout cholangiocarcinoma (HuCCT1) cell line.

Entrez Gene ID	Gene Symbol	Gene Name	Log2 Fold Change	Adjusted *p*-Value
3620	IDO1	indoleamine 2,3-dioxygenase 1	4.88	0.036
2537	IFI6	interferon alpha inducible protein 6	3.41	0.044
4680	CEACAM6	CEA cell adhesion molecule 6	3.05	0.028
91543	RSAD2	radical S-adenosyl methionine domain-containing 2	2.91	0.017
27074	LAMP3	lysosomal associated membrane protein 3	2.90	0.015
4600	MX2	MX dynamin-like GTPase 2	2.84	0.017
4599	MX1	MX dynamin-like GTPase 1	2.79	0.030
4939	OAS2	2’-5’-oligoadenylate synthetase 2	2.67	0.017
8638	OASL	2’-5’-oligoadenylate synthetase-like	2.67	0.014
8519	IFITM1	interferon-induced transmembrane protein 1	2.62	0.040
9636	ISG15	ISG15 ubiquitin-like modifier	2.61	0.008
115361	GBP4	guanylate-binding protein 4	2.57	0.047
3437	IFIT3	interferon-induced protein with tetratricopeptide repeats 3	2.55	0.017
81030	ZBP1	Z-DNA-binding protein 1	2.48	0.011
4938	OAS1	2’-5’-oligoadenylate synthetase 1	2.43	0.008
10406	WFDC2	WAP four-disulfide core domain 2	2.38	0.031
3075	CFH	complement factor H	2.38	0.022
3434	IFIT1	interferon-induced protein with tetratricopeptide repeats 1	2.36	0.017
25984	KRT23	keratin 23	2.31	0.008
3433	IFIT2	interferon-induced protein with tetratricopeptide repeats 2	2.30	0.018

**Table 4 ijms-22-06303-t004:** Relationship between PODXL2 expression and clinicopathological characteristics of intrahepatic cholangiocarcinoma.

		Number of Cases (*n* = 52)	PODXL2 Expression		
			Negative (*n* = 35)	Positive (*n* = 17)	*p*-Value
Gender	Male	39	26	13	0.575
	Female	13	9	4	
Age (mean) (years)			73.9 (39–84)	73.7 (51–81)	0.731
Tumor size (mean) (mm)			66.2 (18–220)	63.9 (20–114)	0.519
Localization	Peripheral	41	31	10	0.0197 *
	Non-peripheral	11	4	7	
Macroscopic	MF	46	33	13	0.0808
type	MF + PI, IG + PI, PI	6	2	4	
Histological	Well	7	3	4	0.0104 *
differentiation	Moderate	32	19	13	
	Poor	13	13	0	
Growth type	Expansive	24	20	4	0.0223 *
	Infiltrative	28	15	13	
Serosa invasion	+	25	13	12	0.0239 *
	−	27	22	5	
Portal vein	+	42	27	15	0.290
invasion	−	10	8	2	
Hepatic vein	+	22	13	9	0.217
invasion	−	30	22	8	
Hepatic artery	+	5	3	2	0.533
invasion	−	47	32	15	
Bile duct	+	27	13	13	0.0038 **
invasion	−	25	22	3	
Intrahepatic	+	21	12	9	0.162
metastasis	−	31	23	8	
Lymph node	+	14	8	6	0.266
metastasis	−	38	27	11	

MF: mass-forming type, PI: periductal-infiltrating type, IG: intraductal-growth type; *, *p* < 0.05; **, *p* < 0.01.

## Data Availability

The data presented in this study are available in this article or Appendix A.

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
