# Peer review of "ITGB6-Knockout Suppresses Cholangiocarcinoma Cell Migration and Invasion with Declining PODXL2 Expression"

_ijms, 2021, doi:10.3390/ijms22126303_

Round 1

Reviewer 1 Report

The authors have generated two cell clones derived from an iCCA cell line, HuCCT1, to study the role of Integrin avb6 in cholangiocarcinoma in vitro model, and they have described that down-regulation of the protein perturbed the cell cycle distribution and decreased the migration and invasion potential. .

However, the role of Integrin avb6 has already been studied in cholangiocarcinoma cell lines (iCCA RBE cells and eCCA QBC939 eCCA cells) and it has already been shown that its silencing by specific siRNA produced the same effect as the authors mentioned in discussion. The novelty is the study of the modulation of genes and pathways in an ITGB6 knockout clone, but the work need some major revisions

Major revisions

1) The major limitation of the study is that only one in vitro model was used although they investigated the biological role of Integrin avb6 in two ITGB6 knockout clones.

2) I would not define clones knockout, but knock-down, given that the expression of the protein is only reduced and not completely inhibited.

3) The cell cycle and invasive and migration potential in the two clones were evaluated. It would be useful to verify cell growth and apoptosis.

4) The gene modulation determined by RNA seq, following ITGB6 knockout, was investigated on a single clone and validated in real- time on the other clone. However, comparing the gene profile (RNA seq) of two different cell line models would be useful in identifying a common panel of deregulated genes. Furthermore, of the validated genes, only PODXL2 (down-regulated) and CEACAM6 (up-regulated) are concordant in the two clones. The authors do not give any explanations about this issue.

5) Why was only the most down-regulated gene ingested and not one of the up-regulated genes? Authors should justify the choice.

Minor revision

4) The table of the top 20 upregulated genes should not be a supplementary Table S2, but a table in the text.

Author Response

We wish to express our appreciation to the reviewers for their insightful comments on our paper. The comments helped us to improve the paper. Please see the attachment.

Reviewer 2 Report

The present manuscript by Soejima et al. describes the successful creation of human knockout cell lines using Crispr-Cas technology to study the molecular background of cholangiocarcinoma. Knockout results in strong downregulation of podocalyxin like protein 2.

The information supplied has been well written and data are presented overall in a clear manner. Furthermore, data are overall well substantiated by statistics. As said, the conclusion on association of integrin β6 and podocalyxin like protein 2 is very interesting.

The following issues should be taken into consideration before publication of this manuscript.

Abstract:

Line 15: Although I would be careful using too many abbreviations in this section, I would write the acronym CRISPR directly after ‘clustered regulatory… repeats’ as well as the abbreviation Cas after ‘associated nuclease 9’. It gives emphasis on the technology and raises the attention immediately to the presented text.

Line 22: podocalyxin like 2: should be podocalyxin like protein 2.

Introduction:

Line 56: maybe start with ‘Studies from our laboratory have demonstrated the association of integrins with iCCA’ or likewise similar text (‘Previous studies have suggested a role for…’), followed by some general information on the integrins.

Line 82: the additional text ‘including cell migration… proliferation’ is redundant. These are general features of carcinogenic cells.

Results:

Line 198: RNA seq was performed on KO cell line 1. Since KO 1 and 2 do not completely behave identical, as is demonstrated by some experiments (sequences were not identical either), it would be informative to give the rationale to pursue with knockout cell line 1.

Lines 217 and further, figure 4: it appears to me that Claudin2 does show a significant change compared to wild type. In addition, there is a clear difference between the two KO cell lines. The text in the legend states that there is no difference. Can you explain this more? This point also relates to the previous issue of choosing KO cell line 1. And, in general, can we compare the two KO cell lines with each other and use them both as models for the KO situation? For S100A2, the expression is lower for KO1 than for the wild type, which is also significant (not stated as such in the legend text).

Line 261: the expression of CD56 (red). All images demonstrate blue color; some KO1 cells stain red. Is this what the reader should deduce from the information? And, the difference is significant, according to the legend text. Consider rewriting the legend in this aspect.

Discussion:

Line 267: the knockout cell line 2 has a large deletion; this was sequenced and mentioned as such, which is fine. It worries me a bit that these KO lines do not behave identical in all experiments. Can there be more off target effects of the guide RNA used? Has this been checked before in silico? I do not read that in the materials and methods section.

Materials and methods:

Line 367: primers for expressed genes are presented in the supplementary file, but I miss the primers that were used for PCR amplification after transfection. The off target analysis was already mentioned under the previous point (which software was used?). Last, three gRNA’s are presented, but one yielded KO clones. Maybe spend a sentence on the different gRNA’s and the resulting clones in the results section (in paragraph 2.1).

Minor issues:

Line 71: Typo cholangiolocellular

Line 199: 60 728 genes. Remove space; add komma 60,728

Line 316: ‘The possible relationship …. ‘. I would suggest to make the phrase stronger and avoiding the past tense (‘was described’), like: In the present study we demonstrate a possible relationship between PODXL2 and integrin β6. The issue returns in line 332.

Author Response

(The authors gave the same response as above.)

Round 2

Reviewer 1 Report

Dear authors,

some answers are not satisfactory, in particular I believe that the analyses of cell growth and apoptosis (at least in in vitro studies) are important to be able to argue that ITGB6 could be a potential therapeutic target (as written in abstract and discussion)

Thank you

Author Response

Thank you very much for your kind and thoughtful comments. I would like to answer your comments. Please see the attachment.

Round 3

Reviewer 1 Report

The paper in now acceptable